# Minimality Conditions Equivalent to the Finitude of Fermat and Mersenne Primes

## Menachem Shlossberg

School of Computer Science, Reichman University, 4610101 Herzliya, Israel; menachem.shlossberg@post.runi.ac.il

**Abstract:** The question is still open as to whether there exist infinitely many Fermat primes or infinitely many composite Fermat numbers. The same question concerning Mersenne numbers is also unanswered. Extending some recent results of Megrelishvili and the author, we characterize the Fermat primes and the Mersenne primes in terms of the topological minimality of some matrix groups. This is achieved by showing, among other things, that if $\mathbb{F}$ is a subfield of a local field of characteristic $\neq 2$, then the special upper triangular group $\mathrm{ST}^+(n, \mathbb{F})$ is minimal precisely when the special linear group $\mathrm{SL}(n, \mathbb{F})$ is. We provide criteria for the minimality (and total minimality) of $\mathrm{SL}(n, \mathbb{F})$ and $\mathrm{ST}^+(n, \mathbb{F})$, where $\mathbb{F}$ is a subfield of $\mathbb{C}$. Let $\mathcal{F}_\pi$ and $\mathcal{F}_c$ be the set of Fermat primes and the set of composite Fermat numbers, respectively. As our main result, we prove that the following conditions are equivalent for $\mathcal{A} \in \{\mathcal{F}_\pi, \mathcal{F}_c\}$: $\mathcal{A}$ is finite; $\prod_{F_n \in \mathcal{A}} \mathrm{SL}(F_n - 1, \mathbb{Q}(i))$ is minimal, where $\mathbb{Q}(i)$ is the Gaussian rational field; and $\prod_{F_n \in \mathcal{A}} \mathrm{ST}^+(F_n - 1, \mathbb{Q}(i))$ is minimal. Similarly, denote by $\mathcal{M}_\pi$ and $\mathcal{M}_c$ the set of Mersenne primes and the set of composite Mersenne numbers, respectively, and let $\mathcal{B} \in \{\mathcal{M}_\pi, \mathcal{M}_c\}$. Then the following conditions are equivalent: $\mathcal{B}$ is finite; $\prod_{M_p \in \mathcal{B}} \mathrm{SL}(M_p + 1, \mathbb{Q}(i))$ is minimal; and $\prod_{M_p \in \mathcal{B}} \mathrm{ST}^+(M_p + 1, \mathbb{Q}(i))$ is minimal.

**Keywords:** Fermat primes; Fermat numbers; Mersenne primes; minimal group; special linear group; Gaussian rational field

**MSC:** 11A41; 11Sxx; 20H20; 54H11; 54H13

## 1. Introduction

A Fermat number has the form $F_n = 2^{2^n} + 1$, where $n$ is a non-negative integer while a Mersenne number has the form $M_p = 2^p - 1$ for some prime $p$. Note that $2^n - 1$ is composite when $n$ is composite. In other words, a Mersenne prime is a prime number that is one less than a power of two. There are several open problems concerning these numbers (e.g., see [1]). For example, it is still unknown whether there are infinitely many Fermat primes, composite Fermat numbers, Mersenne primes, or composite Mersenne numbers.

All topological groups in this paper are Hausdorff. Let $\mathbb{F}$ be a topological subfield of a local field. Recall that a *local field* is a non-discrete locally compact topological field. Denote by $\mathrm{SL}(n, \mathbb{F})$ the special linear group over $\mathbb{F}$ of degree $n$ equipped with the pointwise topology inherited from $\mathbb{F}^{n^2}$, and by $\mathrm{ST}^+(n, \mathbb{F})$ its topological subgroup consisting of upper triangular matrices. In [2], Megrelishvili and the author characterized Fermat primes in terms of the topological minimality of some special linear groups. Recall that a topological group $G$ is *minimal* [3,4] if every continuous isomorphism $f \colon G \to H$, with $H$ a topological group, is a topological isomorphism (equivalently, if $G$ does not admit a strictly coarser Hausdorff group topology).

**Theorem 1** ( [2], Theorem 5.5). *For an odd prime $p$, the following conditions are equivalent:*

*1. $p$ is a Fermat prime;*

2.  SL$(p - 1, (\mathbb{Q}, \tau_p))$ *is minimal, where* $(\mathbb{Q}, \tau_p)$ *is the field of rationals equipped with the p-adic topology;*

3.  SL$(p - 1, \mathbb{Q}(i))$ *is minimal, where* $\mathbb{Q}(i) \subset \mathbb{C}$ *is the Gaussian rational field.*

A similar characterization of Mersenne primes is provided in Theorem 5. Note that it follows from Gauss–Wantzel Theorem that an odd prime $p$ is a Fermat prime if and only if a $p$-sided regular polygon can be constructed with compass and straightedge.

We prove in Theorem 2 that if $\mathbb{F}$ is a subfield of a local field of characteristics distinct from 2, then the special upper triangular group ST$^+(n, \mathbb{F})$ is minimal if and only if the special linear group SL$(n, \mathbb{F})$ is minimal. This result with some other tools yields criteria for the minimality (and total minimality) of SL$(n, \mathbb{F})$ and ST$^+(n, \mathbb{F})$, where $\mathbb{F}$ is a subfield of $\mathbb{C}$ (see Proposition 2, Remark 2, and Corollary 1).

As a main result, we prove in Theorem 8 that the finitude of Fermat and Mersenne primes as well as the finitude of composite Fermat and Mersenne numbers is equivalent to the minimality of some topological products of some matrix groups.

**2. Minimality of ST$^+(n, \mathbb{F})$ and SL$(n, \mathbb{F})$**

Let $\mathsf{N} := \mathrm{UT}(n, \mathbb{F})$ and $\mathsf{A}$ be the subgroups of ST$^+(n, \mathbb{F})$ consisting of upper unitriangular matrices and diagonal matrices, respectively. Note that $\mathsf{N}$ is normal in ST$^+(n, \mathbb{F})$ and ST$^+(n, \mathbb{F}) \cong \mathsf{N} \rtimes_\alpha \mathsf{A}$, where $\alpha$ is the action by conjugations. It is known that $\mathsf{N}$ is the derived subgroup of ST$^+(n, \mathbb{F})$. Recall also that SL$(n, \mathbb{F})$ has finite center (e.g., see [5] (3.2.6)).

$$Z(\mathrm{SL}(n, \mathbb{F})) = \{\lambda I_n : \lambda \in \mu_n\},$$

where $\mu_n$ is a finite group consisting of the $n$-th roots of unity in $\mathbb{F}$ and $I_n$ is the identity matrix of size $n$.

**Lemma 1.** *Let* $\mathbb{F}$ *be a field and* $n \in \mathbb{N}$*. Then,* $Z(\mathrm{ST}^+(n, \mathbb{F})) = Z(\mathrm{SL}(n, \mathbb{F}))$ *and* ST$^+(n, \mathbb{F})/Z(\mathrm{ST}^+(n, \mathbb{F}))$ *is center-free.*

**Proof.** Let $(C, D) \in Z(\mathsf{N} \rtimes \mathsf{A})$ and $E \in \mathsf{A}$ such that $e_{ii} \neq e_{jj}$ whenever $i \neq j$. For every $i > j$ it holds that

$$(CE)_{ij} = \sum_{t=1}^n c_{it}e_{tj} = c_{ij}e_{jj}$$

and

$$(EC)_{ij} = \sum_{t=1}^n e_{it}c_{tj} = c_{ij}e_{ii}$$

Then, the equality $(C, D)(I, E) = (I, E)(C, D)$ implies that $CE = EC$ and $c_{ij}e_{jj} = c_{ij}e_{ii}$. As $e_{ii} \neq e_{jj}$, we deduce that $c_{ij} = 0$. Since $C$ is an upper unitriangular matrix, it follows that $C = I$. To prove that $Z(\mathrm{ST}^+(n, \mathbb{F})) = Z(\mathrm{SL}(n, \mathbb{F}))$, it suffices to show that the diagonal matrix $D$ is scalar. To this aim, pick distinct indices $i, j$ and a matrix $F \in \mathsf{N}$ such that $f_{ij} \neq 0$. As $(I, D) \in Z(\mathsf{N} \rtimes \mathsf{A})$, it follows that $(I, D)(F, I) = (F, I)(I, D)$. This implies that $DF = FD$ and, in particular, $(DF)_{ij} = (FD)_{ij}$. This yields the equality $d_{ii}f_{ij} = d_{jj}f_{ij}$ since $D$ is diagonal. We conclude that $d_{ii} = d_{jj}$ in view of the inequality $f_{ij} \neq 0$. This proves that $Z(\mathrm{ST}^+(n, \mathbb{F})) = Z(\mathrm{SL}(n, \mathbb{F}))$.

Now, let $(B, D)Z(\mathsf{N} \rtimes \mathsf{A}) \in Z(\mathsf{N} \rtimes \mathsf{A}/Z(\mathsf{N} \rtimes \mathsf{A}))$ and $(C, E) \in \mathsf{N} \rtimes \mathsf{A}$. By what we proved, there exists a scalar $\lambda \in \mathbb{F}$ such that $(B, D)(C, E) = (C, E)(B, D)(I, \lambda I)$. Therefore, $DE = \lambda DE$ and $\lambda = 1$. This means that $(B, D)(C, E) = (C, E)(B, D)$ for every $(C, E) \in \mathsf{N} \rtimes \mathsf{A}$. Therefore, $(\mathsf{N} \rtimes \mathsf{A})/Z(\mathsf{N} \rtimes \mathsf{A})$ and its isomorphic copy ST$^+(n, \mathbb{F})/Z(\mathrm{ST}^+(n, \mathbb{F}))$ are center-free. $\square$

The following lemma will be useful in proving Theorem 2.

**Lemma 2.** *Let $\mathbb{F}$ be a subfield of a field $H$ and let $n \geq 3$ be a natural number. If $L$ is a normal subgroup of $\mathrm{ST}^+(n, H)$ that intersects $\mathrm{UT}(n, H)$ non-trivially, then it intersects $\mathrm{UT}(n, \mathbb{F})$ non-trivially.*

**Proof.** Since $L \cap \mathrm{UT}(n, H)$ is a non-trivial normal subgroup of the nilpotent group $\mathrm{UT}(n, H)$, it must non-trivially intersect the center $Z(\mathrm{UT}(n, H))$. Then there exists

$$
I \neq B = \begin{pmatrix} 1 & 0 & \ldots & 0 & b \\ 0 & 1 & \ddots & \vdots & 0 \\ \vdots & \ddots & \ddots & \ddots & \vdots \\ \vdots & \ldots & 0 & 1 & 0 \\ 0 & \ldots & \ldots & 0 & 1 \end{pmatrix} \in L \cap Z(\mathrm{UT}(n, H))
$$

for some $b \in H$ (see [6] (p. 94) for example). Since $n \geq 3$ there exists a diagonal matrix $D \in \mathrm{ST}^+(n, H)$ such that $d_{11} = b^{-1}$ and $d_{nn} = 1$. This implies that

$$
I \neq DBD^{-1} \in Z(\mathrm{UT}(n, \mathbb{F})). \quad \square
$$

**Definition 1.** *Let $H$ be a subgroup of a topological group $G$. Then $H$ is essential in $G$ if $H \cap L \neq \{e\}$ for every non-trivial closed normal subgroup $L$ of $G$.*

The following minimality criterion of dense subgroups is well known (for compact $G$ see also [4,7]).

**Fact 1.** *Let $H$ be a dense subgroup of a topological group $G$. Then, $H$ is minimal if and only if $G$ is minimal and $H$ is essential in $G$ [8] (minimality criterion).*

**Remark 1.** *If $\mathbb{F}$ is a subfield of a local field $P$, then its completion $\widehat{\mathbb{F}}$ is a topological field that can be identified with the closure of $\mathbb{F}$ in $P$. In case $\mathbb{F}$ is infinite, then $\widehat{\mathbb{F}}$ is also a local field, as the local field $P$ contains no infinite discrete subfields (see [9] (p. 27)).*

**Proposition 1** ([2], Proposition 5.1). *Let $\mathbb{F}$ be a subfield of a local field. Then the following conditions are equivalent:*

1.  $\mathrm{SL}(n, \mathbb{F})$ *is minimal;*
2.  *Any non-trivial central subgroup of $\mathrm{SL}(n, \widehat{\mathbb{F}})$ intersects $\mathrm{SL}(n, \mathbb{F})$ non-trivially (i.e., if $1 \neq \lambda \in \mu_n(\widehat{\mathbb{F}})$, then there exists $k \in \mathbb{Z}$ such that $1 \neq \lambda^k \in \mathbb{F}$).*

**Theorem 2.** *Let $\mathbb{F}$ be a subfield of a local field of characteristic distinct from 2. Then, $\mathrm{SL}(n, \mathbb{F})$ is minimal if and only if $\mathrm{ST}^+(n, \mathbb{F})$ is minimal.*

**Proof.** Without a loss of generality, we may assume that $\mathbb{F}$ is infinite. Suppose first that $\mathrm{ST}^+(n, \mathbb{F})$ is minimal. By Lemma 1, $Z(\mathrm{ST}^+(n, \mathbb{F})) = Z(\mathrm{SL}(n, \mathbb{F}))$. Since this center is finite it follows from the minimality criterion that any non-trivial central subgroup of $\mathrm{ST}^+(n, \widehat{\mathbb{F}})$ intersects $\mathrm{ST}^+(n, \mathbb{F})$ non-trivially. This implies that any non-trivial central subgroup of $\mathrm{SL}(n, \widehat{\mathbb{F}})$ intersects $\mathrm{SL}(n, \mathbb{F})$ non-trivially. By Proposition 1, $\mathrm{SL}(n, \mathbb{F})$ is minimal.

Conversely, let us assume that $\mathrm{SL}(n, \mathbb{F})$ is minimal. In case $n = 2$, then $\mathrm{ST}^+(n, \mathbb{F})$ is minimal by [2] (Theorem 3.4) as an infinite subfield of a local field is locally retrobounded and non-discrete. So, we may assume that $n \geq 3$. By [2] (Theorem 3.19), $\mathrm{ST}^+(n, \widehat{\mathbb{F}})$ is minimal as $\widehat{\mathbb{F}}$ is a local field (see Remark 1). In view of the minimality criterion, it suffices to show that $\mathrm{ST}^+(n, \mathbb{F})$ is essential in $\mathrm{ST}^+(n, \widehat{\mathbb{F}})$. Let $L$ be a closed normal non-trivial subgroup of $\mathrm{ST}^+(n, \widehat{\mathbb{F}})$. If

$$
L \subseteq Z(\mathrm{ST}^+(n, \widehat{\mathbb{F}})) = Z(\mathrm{SL}(n, \widehat{\mathbb{F}})),
$$

then $L$ intersects $\mathrm{SL}(n, \mathbb{F})$ non-trivially by Proposition 1. Clearly, this implies that $L$ intersects $\mathrm{ST}^+(n, \mathbb{F})$ non-trivially. If $L$ is not central, then it must non-trivially intersect

$UT(n, \widehat{\mathbb{F}})$, the derived subgroup of $ST^+(n, \widehat{\mathbb{F}})$, in view of [10] (Lemma 2.3). Now, Lemma 2 implies that $L$ intersects $ST^+(n, \mathbb{F})$ non-trivially and we deduce that $ST^+(n, \mathbb{F})$ is essential in $ST^+(n, \widehat{\mathbb{F}})$. □

In view of Theorems 1 and 2, the following characterization of Fermat primes is obtained.

**Theorem 3.** *For an odd prime p the following conditions are equivalent:*

1. *p is a Fermat prime;*
2. $ST^+(p-1, (\mathbb{Q}, \tau_p))$ *is minimal;*
3. $ST^+(p-1, \mathbb{Q}(i))$ *is minimal.*

The following concept has a key role in the total minimality criterion.

**Definition 2.** *A subgroup H of a topological group G is totally dense if for every closed normal subgroup L of G the intersection $L \cap H$ is dense in L.*

**Fact 2** ([11], total minimality criterion)**.** *Let H be a dense subgroup of a topological group G. Then, H is totally minimal if and only if G is totally minimal and H is totally dense in G.*

**Theorem 4** ([2], Theorem 4.7)**.** *Let $\mathbb{F}$ be a subfield of a local field. Then $SL(n, \mathbb{F})$ is totally minimal if and only if $Z(SL(n, \mathbb{F})) = Z(SL(n, \widehat{\mathbb{F}}))$ (i.e., $\mu_n(\mathbb{F}) = \mu_n(\widehat{\mathbb{F}})$).*

Let $\rho_m = e^{\frac{2\pi i}{m}}$ be the $m$-th primitive root of unity. The next result extends [2] (Corollary 5.3), where $\rho_4 = i$ is considered.

**Proposition 2.** *Let $\mathbb{F}$ be a dense subfield of $\mathbb{C}$. Then,*

1. $SL(n, \mathbb{F})$ *is totally minimal if and only if $\rho_n \in \mathbb{F}$;*
2. $SL(n, \mathbb{F})$ *is minimal if and only if $\langle \rho_m \rangle \cap \mathbb{F}$ is non-trivial whenever m divides n.*

**Proof.** (1) Necessity: Follows from Theorem 4. Indeed, $\lambda = \rho_n \in \mathbb{C}$ is an $n$-th root of unity. Sufficiency: If $\lambda \in \mathbb{C}$ and $\lambda^n = 1$, then $\lambda \in \langle \rho_n \rangle \subseteq \mathbb{F}$. So, we may use Theorem 4 again.

(2) Necessity: Let $1 \neq \lambda \in \mathbb{C}$ be an $n$-th root of unity. Then, $\lambda$ is an $m$-th primitive root of unity where $m$ divides $n$. Since $SL(n, \mathbb{F})$ is minimal, it follows that there exists $k$ such that $1 \neq \lambda^k \in \mathbb{F}$. Clearly, $\lambda^k \in \langle \rho_m \rangle \cap \mathbb{F}$. So, $\langle \rho_m \rangle \cap \mathbb{F}$ is non-trivial. Now use Proposition 1.

Sufficiency: Let $1 \neq \lambda \in \mathbb{C}$ be an $n$-th root of unity. Then, $\lambda$ is an $m$-th primitive root of unity where $m$ divides $n$. This means that $\lambda = e^{\frac{2\pi i k}{m}} = (\rho_m)^k$, where $1 \leq k \leq m$ with $\gcd(k, m) = 1$. By our assumption, $\langle \rho_m \rangle \cap \mathbb{F}$ is non-trivial. Hence, there exists $l$ such that $1 \neq (\rho_m)^l \in \langle \rho_m \rangle \cap \mathbb{F}$. Since $\gcd(k, m) = 1$ and $\lambda = (\rho_m)^k$, it follows that there exists $t \in \mathbb{Z}$ such that $(\rho_m)^l = \lambda^t$. This proves the minimality of $SL(n, \mathbb{F})$, in view of Proposition 1. □

**Remark 2.** *It is known that a subfield $\mathbb{F}$ is dense in $\mathbb{C}$ if and only if it is not contained in $\mathbb{R}$. By [2] (Corollary 4.8), if $\mathbb{F} \subseteq \mathbb{R}$, then $SL(n, \mathbb{F})$ is totally minimal for every $n \in \mathbb{N}$. So, together with Proposition 2, we obtain criteria for the minimality and total minimality of $SL(n, \mathbb{F})$, where $\mathbb{F}$ is any subfield of $\mathbb{C}$ and $n \in \mathbb{N}$.*

Since $\mathbb{C}$ has zero characteristic, Theorem 2, Proposition 2, and Remark 2 imply the following:

**Corollary 1.** *Let $\mathbb{F}$ be a topological subfield of $\mathbb{C}$.*

1. *If $\mathbb{F}$ is dense in $\mathbb{C}$, then $ST^+(n, \mathbb{F})$ is minimal if and only if $\langle \rho_m \rangle \cap \mathbb{F}$ is non-trivial whenever m divides n.*
2. *If $\mathbb{F} \subseteq \mathbb{R}$, then $ST^+(n, \mathbb{F})$ is minimal for every $n \in \mathbb{N}$.*

### 3. Proof of the Main Result

By [2] (Corollary 5.3), $\mathrm{SL}(n, \mathbb{Q}(i))$ is minimal if and only if $n = 2^k$, where $k$ is a non-negative integer. This immediately implies the following theorem concerning Mersenne primes (compare with Theorems 1 and 3).

**Theorem 5.** *For a prime p the following conditions are equivalent:*

1. *p is a Mersenne prime;*
2. *$\mathrm{SL}(p + 1, \mathbb{Q}(i))$ is minimal;*
3. *$\mathrm{ST}^+(p + 1, \mathbb{Q}(i))$ is minimal.*

At this point, one may expect to have similar characterizations of the Mersenne primes involving the $p$-adic topology (see item 2 of Theorem 3). Nevertheless, the following proposition holds for all primes and not just for the Mersenne primes.

**Proposition 3.** *Let $\mathbb{F}$ be a topological subfield of $\mathbb{Q}_p$, where p is a prime number.*

1. *$\mathrm{SL}(p + 1, \mathbb{F})$ is totally minimal.*
2. *$\mathrm{ST}^+(p + 1, \mathbb{F})$ is minimal.*

**Proof.** (1) By [2] (Corollary 4.8), it suffices to show that

$$Z(\mathrm{SL}(p + 1, \mathbb{Q}_p)) = \{I, -I\}.$$

It is known that $\pm 1$ are the only roots of unity in $\mathbb{Q}_2$ and that for $p > 2$ the roots of unity in $\mathbb{Q}_p$ form a cyclic group of order $p - 1$ (see [12] (p. 15)). So, the assertion holds for $p = 2$. Now assume that $p > 2$ and $\lambda^{p+1} = 1$. On the one hand, the order of $\lambda$ must divide $p - 1$ as $\lambda$ is a root of unity. On the other hand, we must also have $o(\lambda) | (p + 1)$. Since $2 = p + 1 - (p - 1)$, it follows that $o(\lambda) | 2$ and we deduce that $Z(\mathrm{SL}(p + 1, \mathbb{Q}_p)) = \{I, -I\}$.
(2) By (1), $\mathrm{SL}(p + 1, \mathbb{F})$ is minimal. In view of Theorem 2, $\mathrm{ST}^+(p + 1, \mathbb{F})$ is also minimal. □

In the sequel, we will always equip a product of topological groups with the product topology.

**Theorem 6.**

1. *If $\mathbb{F}$ is a local field, then $\prod_{n \in \mathbb{N}} \mathrm{SL}(n, \mathbb{F})$ is minimal.*
2. *If, in addition, $\mathrm{char}(\mathbb{F}) \neq 2$, then $\prod_{n \in \mathbb{N}} \mathrm{ST}^+(n, \mathbb{F})$ is minimal.*

**Proof.** (1) Since a compact group is minimal, we may assume without loss of generality that $\mathbb{F}$ is infinite. By [13] (see also [2] (Theorem 4.3)), the projective special linear group $\mathrm{PSL}(n, \mathbb{F}) = \mathrm{SL}(n, \mathbb{F})/Z(\mathrm{SL}(n, \mathbb{F}))$ (equipped with the quotient topology) is minimal for every $n \in \mathbb{N}$. Being algebraically simple (see [5] (3.2.9)), $\mathrm{PSL}(n, \mathbb{F})$ has a trivial center. Therefore, the topological product $\prod_{n \in \mathbb{N}} \mathrm{PSL}(n, \mathbb{F})$ is minimal by [14] (Theorem 1.15). As,

$$\prod_{n \in \mathbb{N}} \mathrm{PSL}(n, \mathbb{F}) \cong \prod_{n \in \mathbb{N}} \mathrm{SL}(n, \mathbb{F}) / Z\left(\prod_{n \in \mathbb{N}} \mathrm{SL}(n, \mathbb{F})\right),$$

where $Z\left(\prod_{n \in \mathbb{N}} \mathrm{SL}(n, \mathbb{F})\right)$ is compact, it follows from [15] (Theorem 7.3.1) that $\prod_{n \in \mathbb{N}} \mathrm{SL}(n, \mathbb{F})$ is minimal.
(2) By Lemma 1, the center of $\mathrm{ST}^+(n, \mathbb{F})/Z(\mathrm{ST}^+(n, \mathbb{F}))$ is trivial for every $n \in \mathbb{N}$, where $Z(\mathrm{ST}^+(n, \mathbb{F})) = Z(\mathrm{SL}(n, \mathbb{F}))$. By [2] (Theorem 3.17), $\mathrm{ST}^+(n, \mathbb{F})/Z(\mathrm{ST}^+(n, \mathbb{F}))$ is minimal. We complete the proof using the topological isomorphism

$$\prod_{n \in \mathbb{N}} \mathrm{ST}^+(n, \mathbb{F}) / \prod_{n \in \mathbb{N}} Z(\mathrm{ST}^+(n, \mathbb{F})) \cong \prod_{n \in \mathbb{N}} (\mathrm{ST}^+(n, \mathbb{F})/Z(\mathrm{ST}^+(n, \mathbb{F})))$$

and similar arguments to those appearing in the proof of (1). □

**Remark 3.** *In their recent paper [16], the authors call a minimal group G z-minimal if $G/Z(G)$ is minimal. In view of the results obtained in [2,8], it holds that in case $\mathbb{F}$ is a local field, then $\mathrm{SL}(n, \mathbb{F})$ is z-minimal. Moreover, by Lemma 1 and [2] (Theorem 3.17) also $\mathrm{ST}^+(n, \mathbb{F})$ is z-minimal in case the local field $\mathbb{F}$ has a characteristic distinct from $2$. By [16] (Corollary 4.9), a product of complete z-minimal groups is minimal. This provides an alternative proof for Theorem 6.*

**Definition 3.** *[17] A minimal group G is perfectly minimal if $G \times H$ is minimal for every minimal group H.*

**Proposition 4.** *Let $\mathbb{F}$ be a subfield of a local field. Then $\mathrm{SL}(2^k, \mathbb{F})$ is perfectly minimal for every $k \in \mathbb{N}$. If $\mathrm{char}(\mathbb{F}) \neq 2$, then $\mathrm{ST}^+(2^k, \mathbb{F})$ is perfectly minimal for every $k \in \mathbb{N}$.*

**Proof.** Let $\mathbb{F}$ be a subfield of a local field and $k \in \mathbb{N}$. By [2] (Corollary 5.2), $\mathrm{SL}(2^k, \mathbb{F})$ is minimal. Being finite, the center $Z(\mathrm{SL}(2^k, \mathbb{F}))$ is perfectly minimal (see [3]). Having a perfectly minimal center, the minimal group $\mathrm{SL}(2^k, \mathbb{F})$ is perfectly minimal in view of [14] (Theorem 1.4). The last assertion is proved similarly, taking into account that $Z(\mathrm{SL}(2^k, \mathbb{F})) = Z(\mathrm{ST}^+(2^k, \mathbb{F}))$ and the fact that $\mathrm{ST}^+(2^k, \mathbb{F})$ is minimal by Theorem 2. □

**Corollary 2.** *Let $n$ be a non-negative integer and $F_n = 2^{2^n} + 1$ be a Fermat number. Then $\mathrm{SL}(F_n - 1, \mathbb{Q}(i))$ and $\mathrm{ST}^+(F_n - 1, \mathbb{Q}(i))$ are perfectly minimal. If $p = F_n$ is a Fermat prime, then $\mathrm{SL}(p - 1, (\mathbb{Q}, \tau_p))$ and $\mathrm{ST}^+(p - 1, (\mathbb{Q}, \tau_p))$ are perfectly minimal, where $\tau_p$ is the p-adic topology.*

**Proof.** Use Proposition 4 and the fact that $F_n - 1$ is a power of two. □

Recall that if $M_p$ is a Mersenne number then $M_p + 1$ is a power of two. So, we also obtain the following result:

**Corollary 3.** *Let $p$ be a prime number and $M_p = 2^p - 1$ be a Mersenne number. Then $\mathrm{SL}(M_p + 1, \mathbb{Q}(i))$ and $\mathrm{ST}^+(M_p + 1, \mathbb{Q}(i))$ are perfectly minimal.*

Using Proposition 3, together with the arguments appearing in the proof of Proposition 4, one may obtain the following result.

**Corollary 4.** *If $\mathbb{F}$ is a topological subfield of $\mathbb{Q}_p$, where $p$ is a prime number, then both topological groups $\mathrm{SL}(p + 1, \mathbb{F})$ and $\mathrm{ST}^+(p + 1, \mathbb{F})$ are perfectly minimal.*

**Theorem 7.** *Let $(n_k)_{k \in \mathbb{N}}$ be an increasing sequence of natural numbers. Then, neither $\prod_{k \in \mathbb{N}} \mathrm{SL}(2^{n_k}, \mathbb{Q}(i))$ nor $\prod_{k \in \mathbb{N}} \mathrm{ST}^+(2^{n_k}, \mathbb{Q}(i))$ are minimal.*

**Proof.** We first prove that $G = \prod_{k \in \mathbb{N}} \mathrm{SL}(2^{n_k}, \mathbb{Q}(i))$ is not minimal. In view of the minimality criterion, it suffices to show that $G$ is not essential in $\widehat{G} = \prod_{k \in \mathbb{N}} \mathrm{SL}(2^{n_k}, \mathbb{C})$. To this aim, let

$$N = \{(\lambda_k I_{2^{n_k}})_{k \in \mathbb{N}} \in \widehat{G} | \ (\lambda_{k+1})^2 = \lambda_k \ \forall k \in \mathbb{N}\}.$$

The equality $\lambda_{k+1}^2 = \lambda_k$ implies that $N$ is a closed central subgroup of $\widehat{G}$. Moreover, $N$ is non-trivial as the sequence $(n_k)_{k \in \mathbb{N}}$ is increasing. Let us see that $N$ trivially intersects $G$. Otherwise, there exists a sequence $(\lambda_k)_{k \in \mathbb{N}}$ of roots of unity in $\mathbb{Q}(i)$ such that $(\lambda_{k+1})^2 = \lambda_k$ for every $k \in \mathbb{N}$ and $\lambda_{k_0} \neq 1$ for some $k_0 \in \mathbb{N}$. It follows that $\lambda_{k_0}, \lambda_{k_0+1}, \lambda_{k_0+2}, \lambda_{k_0+3}$ are different non-trivial roots of unity in $\mathbb{Q}(i)$, contradicting the fact that $\pm 1, \pm i$ are the only roots of unity in $\mathbb{Q}(i)$.

Now consider the group $H = \prod_{k \in \mathbb{N}} \mathrm{ST}^+(2^{n_k}, \mathbb{Q}(i))$. In view of Lemma 1 and what we have just proved, $N$ is also a closed non-trivial central subgroup of $\widehat{H}$ that trivially intersects $H$. This means that $H$ is not essential in $\widehat{H}$. By the minimality criterion, $H$ is not minimal. □

**Theorem 8.** *1.*     *Let $\mathcal{F}_\pi$ and $\mathcal{F}_c$ be the set of Fermat primes and the set of composite Fermat numbers, respectively, and let $\mathcal{A} \in \{\mathcal{F}_\pi, \mathcal{F}_c\}$. Then, the following conditions are equivalent:*

    *(a)*     $\mathcal{A}$ *is finite;*

    *(b)*     $\prod_{F_n \in \mathcal{A}} \mathrm{SL}(F_n - 1, \mathbb{Q}(i))$ *is minimal;*

    *(c)*     $\prod_{F_n \in \mathcal{A}} \mathrm{ST}^+(F_n - 1, \mathbb{Q}(i))$ *is minimal.*

*2.*     *Let $\mathcal{M}_\pi$ and $\mathcal{M}_c$ be the set of Mersenne primes and the set of composite Mersenne numbers, respectively, and let $\mathcal{B} \in \{\mathcal{M}_\pi, \mathcal{M}_c\}$. Then, the following conditions are equivalent:*

    *(a)*     $\mathcal{B}$ *is finite;*

    *(b)*     $\prod_{M_p \in \mathcal{B}} \mathrm{SL}(M_p + 1, \mathbb{Q}(i))$ *is minimal;*

    *(c)*     $\prod_{M_p \in \mathcal{B}} \mathrm{ST}^+(M_p + 1, \mathbb{Q}(i))$ *is minimal.*

**Proof.** (1) Assume first that $\mathcal{A}$ is finite. It is easy to see that a product of finitely many perfectly minimal groups is minimal. This and Corollary 2 imply that both topological groups $\prod_{F_n \in \mathcal{A}} \mathrm{SL}(F_n - 1, \mathbb{Q}(i))$ and $\prod_{F_n \in \mathcal{A}} \mathrm{ST}^+(F_n - 1, \mathbb{Q}(i))$ are minimal. If $\mathcal{A}$ is infinite, then $\prod_{F_n \in \mathcal{A}} \mathrm{SL}(F_n - 1, \mathbb{Q}(i))$ and $\prod_{F_n \in \mathcal{A}} \mathrm{ST}^+(F_n - 1, \mathbb{Q}(i))$ are not minimal by Theorem 7.

(2) The proof is similar to (1). The only difference is that we use Corollary 3 instead of Corollary 2. $\square$

Due to the fact that there are infinitely many Fermat numbers and infinitely many Mersenne numbers, we immediately obtain the following corollaries of Theorem 8:

**Corollary 5.** *At least one of the following topological products is not minimal:*

- $\prod_{F_n \in \mathcal{F}_\pi} \mathrm{SL}(F_n - 1, \mathbb{Q}(i))$;
- $\prod_{F_n \in \mathcal{F}_c} \mathrm{SL}(F_n - 1, \mathbb{Q}(i))$.

**Corollary 6.** *At least one of the following topological products is not minimal:*

- $\prod_{M_p \in \mathcal{M}_\pi} \mathrm{SL}(M_p + 1, \mathbb{Q}(i))$;
- $\prod_{M_p \in \mathcal{M}_c} \mathrm{SL}(M_p + 1, \mathbb{Q}(i))$.

The next proposition deals with the *p*-adic topology $\tau_p$.

**Proposition 5.** *1.*     *If the set of Fermat primes $\mathcal{F}_\pi$ is finite, then $\prod_{p \in \mathcal{F}_\pi} \mathrm{SL}(p - 1, (\mathbb{Q}, \tau_p))$ and $\prod_{p \in \mathcal{F}_\pi} \mathrm{ST}^+(p - 1, (\mathbb{Q}, \tau_p))$ are minimal.*

*2.*     *If the set of Mersenne primes $\mathcal{M}_\pi$ is finite, then $\prod_{p \in \mathcal{M}_\pi} \mathrm{SL}(p + 1, (\mathbb{Q}, \tau_p))$ and $\prod_{p \in \mathcal{M}_\pi} \mathrm{ST}^+(p + 1, (\mathbb{Q}, \tau_p))$ are minimal.*

**Proof.** As noted above, the product of finitely many perfectly minimal groups is minimal. Using Corollary 2 and Corollary 4, we complete the proof. $\square$

## 4. Open Questions and Concluding Remarks

In view of Proposition 5 and Theorem 8, two natural questions arise.

**Question 4.** *Consider the following conditions:*

*1.*     $\mathcal{F}_\pi$ *is finite;*

*2.*     $\prod_{p \in \mathcal{F}_\pi} \mathrm{SL}(p - 1, (\mathbb{Q}, \tau_p))$ *is minimal;*

*3.*     $\prod_{p \in \mathcal{F}_\pi} \mathrm{ST}^+(p - 1, (\mathbb{Q}, \tau_p))$ *is minimal.*

*Are they equivalent?*

**Question 5.** *Consider the following conditions:*

*1.*     $\mathcal{M}_\pi$ *is finite;*

*2.*     $\prod_{p \in \mathcal{M}_\pi} \mathrm{SL}(p + 1, (\mathbb{Q}, \tau_p))$ *is minimal;*

*3.*     $\prod_{p \in \mathcal{M}_\pi} \mathrm{ST}^+(p + 1, (\mathbb{Q}, \tau_p))$ *is minimal.*

*Are they equivalent?*

Since Proposition 3 deals with all primes, we also ask

**Question 6.** *Let $\mathcal{P}$ be the set of all primes. Are $\prod_{p \in \mathcal{P}} \mathrm{SL}(p+1, (\mathbb{Q}, \tau_p))$ and $\prod_{p \in \mathcal{P}} \mathrm{ST}^+(p+1, (\mathbb{Q}, \tau_p))$ minimal?*

**Remark 7.** *If there were only finitely many primes, then we could have proved that both topological products $\prod_{p \in \mathcal{P}} \mathrm{SL}(p+1, (\mathbb{Q}, \tau_p))$ and $\prod_{p \in \mathcal{P}} \mathrm{ST}^+(p+1, (\mathbb{Q}, \tau_p))$ must be minimal. So, showing that either one of these products is not minimal produces a new topological proof for the infinitude of primes.*

**Funding:** This research received no external funding.

**Institutional Review Board Statement:** Not applicable.

**Informed Consent Statement:** Not applicable.

**Data Availability Statement:** Not applicable.

**Conflicts of Interest:** The authors declare no conflict of interest.

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
