# Peer review of "Minimality Conditions Equivalent to the Finitude of Fermat and Mersenne Primes"

_axioms, doi:10.3390/axioms12060540_

Round 1

Reviewer 1 Report

Review report on “Minimality conditions equivalent to the finitude of Fermat and Mersenne primes”

This paper is concerned with the finitude of Fermat and Mersenne primes. The author characterizes these primes in terms of topological minimality of matrix groups, and shows that the finitude of Fermat and composite Fermat numbers is equivalent to certain conditions involving quadratic forms. The paper provides criteria for the minimality (and total minimality) of certain matrix groups, and extends some results from previous research.

The paper is well-organized and clearly written, with a concise abstract that summarizes the main findings.

Overall, this paper offers useful insights into the relationship between number theory and matrix groups, and sheds new light on the characterization of Fermat and Mersenne primes.

All in all, I recommend the paper to be accepted.

Reviewer 2 Report

Outstanding work! My congratulations on this occasion. 
